# Deciphering the factors influencing electric field mediated polymerization and depolymerization at the solution–solid interface
Zhinan Fu, Nicolás Arisnabarreta, Kunal S. Mali ✉ & Steven De Feyter ✉

Strong and oriented electric fields are known to influence structure as well as reactivity. The strong electric field (EF) between the tip of a scanning tunneling microscope (STM) and graphite has been used to modulate two-dimensional (2D) polymerization of aryl boronic acids where switching the polarity of the substrate bias enabled reversible transition between self-assembled molecular networks of monomers and crystalline 2D polymer (2DP) domains. Here, we untangle the different factors influencing the EF-mediated (de)polymerization of a boroxine-based 2DP on graphite. The influence of the solvent was systematically studied by varying the nature from polar protic to polar aprotic to non-polar. The effect of monomer concentration was also investigated in detail with a special focus on the time-dependence of the transition. Our experimental observations indicate that while the nucleation of 2DP domains is not initiated by the applied electric field, their depolymerization and subsequent desorption, are a consequence of the change in the polarity of the substrate bias within the area scanned by the STM tip. We conclude that the reversible transition is intimately linked to the bias-induced adsorption and desorption of the monomers, which, in turn, could drive changes in the local concentration of the monomers.

Synthetic two-dimensional polymers (2DPs) are structurally precise, ultrathin, sheet-like macromolecules that consist of laterally linked repeat units connected via covalent bonds. Typically obtained using well-defined monomers via dynamic covalent chemistry, 2DPs are promising materials that can expand the realm of 2D materials beyond inorganic systems such as graphene and transition metal dichalcogenides. 2DP sheets, when stacked, yield the so-called 2D covalent organic frameworks (2D-COFs). 2DPs and 2D-COFs have numerous applications in diverse fields including catalysis, separation technology, sensing, and gas storage to name a few[1,2].

There exist different synthetic platforms for obtaining 2D-COFs and 2DPs. 2D-COFs are typically obtained via reversible reactions such as Schiff base formation[3,4], boronic acid self-condensation, and boronate ester formation[5,6]. In recent years, however, alternative chemistries based on Michael addition[7] and Knoevenagel condensation[8] have gained traction for 2D-COF synthesis as they provide relatively stronger linkages. Solvothermal

methods, where the reaction is carried out at high temperature and pressure in an autoclave, are often employed. Such harsh conditions are necessary for maintaining the reversibility of covalent bond formation, which in turn allows the formation of crystalline 2DP sheets which then stack on top of each other. Typically, long reaction times (3–7 days) are needed after which the 2D-COF is isolated as an insoluble microcrystalline powder. The delamination of 2D-COFs into individual monolayers of 2DPs is an insurmountable task that remains a bottleneck in their molecular scale characterization.

As an alternative, especially in the context of nanoscale characterization of the material and understanding the processes that lead up to its formation, other synthetic paradigms are being explored that allow the fabrication and isolation of monolayers or a few layers of 2DPs. These strategies, which can be broadly classified into two groups, employ surfaces as 2D reaction platforms[9]. The first involves the synthesis of 2DPs at fluid

Division of Molecular Imaging and Photonics, Department of Chemistry, Celestijnenlaan 200F, Leuven 3001, Belgium. ✉e-mail: kunal.mali@kuleuven.be; steven.defeyter@kuleuven.be

interfaces where the polymerization reaction is carried out at the air-liquid or liquid-liquid interface. Typically, few-layer polymer sheets are obtained, which can then be transferred to arbitrary substrates. 2DPs fabricated at fluid interfaces have been characterized in detail using transmission electron microscopy[10,11].

The second consists of the synthesis of 2DPs on solid, typically conductive surfaces[12]. The reaction proceeds at the vapor-solid[13,14], vacuum–solid[15,16], or at the solution–solid[17–19] interface often leading to the formation of a monolayer of 2DP which can be characterized at sub-molecular resolution using scanning tunneling microscopy (STM). Given the proximity of a solid surface, the reactions often proceed under relatively mild conditions. Amongst these, synthesis at the liquid-solid interface offers an additional advantage: not only can the final product be imaged, but also the elementary steps involved in the 2D polymerization process namely, nucleation, growth, and ripening, can be monitored in-situ. We have recently reported on the structural and dynamic aspects of boroxine-based 2DPs formed at the solution–graphite interface[18,20]. In these studies, the mechanistic aspects of 2DP formation were studied in-situ, in the "native" growth environment where the polymerization chemistry occurs, in contrast to isolation and subsequent characterization of intermediate products. Such in-situ monitoring allowed the acquisition of qualitative as well as quantitative details such as non-classical modes of ripening, critical nucleation size, nucleation, and growth rates[20]. These studies demonstrate the tremendous potential STM holds for unraveling the mechanistic aspects of 2D polymerization processes.

A peculiar, and almost enigmatic aspect of boroxine polymerization studied using STM, is the electric field (EF) mediated, bias-dependent polymerization and depolymerization process[21–23]. At negative substrate bias, where electrons tunnel from the graphite substrate to the STM tip, the formation of highly ordered single-layered 2DP domains was observed. However, scanning the same area at positive substrate bias, where the direction of electron tunneling is opposite, the 2DP domains underwent depolymerization to yield ordered self-assembled molecular networks

(SAMNs). The 2D polymerization occurred mostly within the scanned area and the monomers outside of the scanned region remained unpolymerized. The observed bond-making and bond-breaking process was ascribed to the strong oriented EF between the tip and the substrate. Given the proximity of the STM tip to the substrate (<1 nm), the static EF at the tunnel junction can reach values as high as $10^9$ volts per meter. This polymerization/depolymerization process, which can be triggered by simply switching the bias, allows real-time monitoring of 2D polymerization at the liquid-solid interface and highlights the significance of the non-thermal activation mechanism.

While the influence of experimental variables such as the choice of monomer concentration[17,18], solvent[18], and substrate[24] on 2D polymerization has been reported in the recent past, these aspects remain unexplored for the EF-mediated (de)polymerization process. Understanding the influence of some of these parameters will provide further insight into the EF-mediated bond-making/breaking processes, which are poorly understood.

In this contribution, we take the next logical steps toward understanding EF-mediated (de)polymerization of boronic acids at the solution–solid interface. The influence of the choice of solvent on the EF-induced (de)polymerization of 1,3,5-*tris*-(4-phenylboronic acid) benzene (TPBA, Fig. 1a) was systematically investigated. STM experiments indicate that the (de)polymerization of boroxine-linked 2DPs occurs in both protic as well as aprotic solvents. This was confirmed by using a pair of solvents with comparable molecular structures. Furthermore, we also discovered that the EF-mediated (de)polymerization process is dependent on the monomer concentration. At higher concentrations, the system oscillates between ordered SAMNs and crystalline 2DP domains, upon switching the polarity of the applied EF from positive to negative, respectively. On the other hand, at lower concentrations, the adsorbed monomers were found to desorb instantly upon the polarity switch from positive to negative. We discuss the plausible reasons behind the observed effect of bias in terms of plausible changes in the local concentration of monomers.

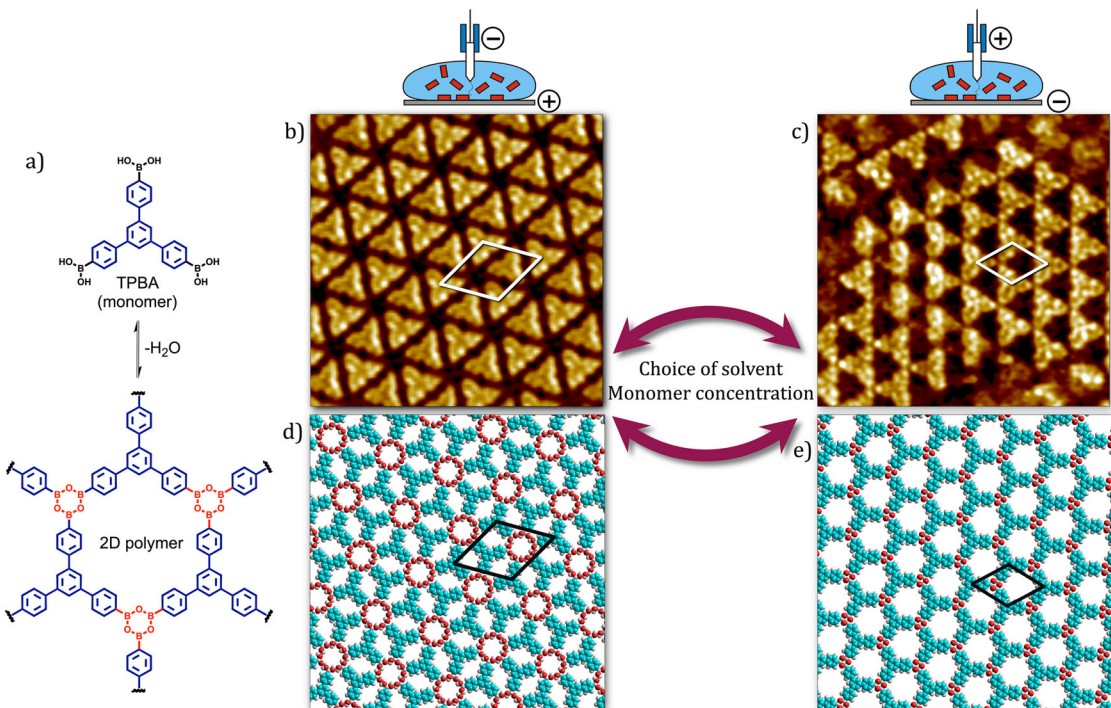

**Fig. 1 | Bias-induced polymerization and depolymerization at the solution–solid interface. a** Schematic showing the 2D polymerization TPBA. STM images depicting the bias-dependent polymerization and depolymerization of TPBA at the OA–graphite interface. At positive substrate bias, a SAMN is formed (**b**, 11 nm × 11 nm) whereas at negative bias, the formation of boroxine-linked 2D polymer (**c**, 11 nm × 11 nm) is observed. The bias-induced (de)polymerization also depends on the choice of the solvent and the concentration of monomers in the solution. Corresponding molecular models for SAMN (**d**) and 2DP (**e**).

**Article**

**Fig. 2 | Bias-induced (de)polymerization of TPBA at the HA–graphite interface. a** SAMN formed at positive sample bias. Imaging conditions: $I_{set}$ = 0.1 nA, $V_{bias}$ = + 0.7 V. White circles highlight the presence of small islands of 2DP in between the SAMN domains. **b** Boroxine-linked 2DP at the negative substrate bias. Imaging conditions: $I_{set}$ = 0.1 nA, $V_{bias}$ = –0.7 V. [TPBA] = 250 µM, scale bar = 20 nm.

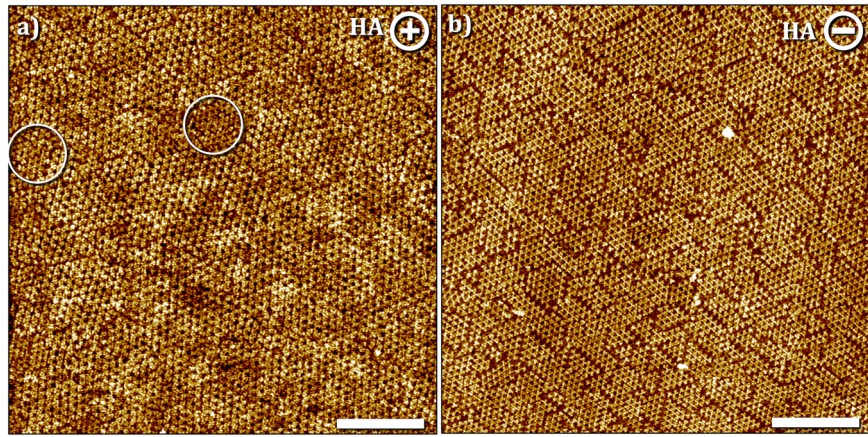

## Results and discussion

In contrast to previous studies, where the on-surface synthesis of boroxine 2DPs was carried out at elevated temperatures[25–29], here we employed a milder synthesis protocol which involves the dissolution of the monomers in a given solvent or solvent mixture and deposition onto the graphite surface at room temperature (RT). The bias-dependent (de)polymerization was then studied at the solution–graphite interface at RT using five different solvents with distinct chemical properties. These solvents include heptanoic acid (HA), octanoic acid (OA), methyl octanoate (MO), 1, 2, 4-trichlorobenzene (TCB), and 1-phenyloctane (1-PO). This series of solvents allows us to probe boroxine 2DP formation and its bias dependence in non-polar (1-PO), polar aprotic (MO, TCB), and polar protic (HA, OA) solvents. The choice of these solvents is further justified by considering their non-volatile nature which is a requirement for in-situ STM experiments. Barring MO, all aforementioned solvents are routinely used in STM experiments under ambient conditions. Considering the anticipated differences in TPBA solubility in these solvents, solid TPBA was first dissolved in dimethyl sulfoxide (DMSO) to ensure complete dissolution of the monomer. The stock solution was then diluted using the aforementioned solvents. We note that the percentage of DMSO in the final solution was between 1% and 2% (v/v), and for alkanoic acids, the experimental results remain the same with and without DMSO. The on-surface synthesis at RT enables real-time monitoring of the (de)polymerization process in situ, providing mechanistic insight.

### EF mediated (de)polymerization: choice of the solvent

Following our previous work[18,20,21], the first set of experiments was carried out using alkanoic acids as solvents. Although boroxines are prone to acid-mediated hydrolytic cleavage, we have reported the formation and STM characterization of boroxine-linked 2DPs at the octanoic acid–graphite interface where the 2DP shows sufficient stability.

Figure 2a shows a representative STM image of the HOPG surface obtained at positive sample bias after deposition of TPBA in heptanoic acid. The approach of the STM tip to the substrate as well as the imaging was started and continued at positive bias in this case. As evident from Fig. 2a,

the monolayer is made up of close-packed units of TPBA. We assign this packing arrangement to a SAMN where the monomers are held together via non-covalent interactions and are arranged in a hexameric fashion with the boronic acid units pointing towards the center of the hexagon. It is plausible that there exist hydrogen bonding interactions between the boronic acid units[30] of adjacent monomers in this packing, which will be referred to as SAMN hereon. We note that the formation of domains of 2DP was also observed in initial scans at positive bias (see Fig. S1 in the supplementary information) which were found to "dissolve" and get removed from the surface in subsequent scans. The smaller domains of 2DP which survived scanning at positive bias can be seen in Fig. 2a (white circles). This indicates that the 2DP nucleates even in the absence of an electric field at the HA–graphite interface which is in line with previous observations[20,23].

Switching the bias to negative and subsequent scanning at the same bias led to a transition (typically 3 min, vide infra) wherein the TPBA monomers underwent polymerization at the HA–graphite interface. Figure 2b shows a representative STM image of the boroxine 2DP obtained at negative values of substrate bias. A network with the anticipated hexagonal symmetry can be readily identified from the STM data. The domain size is typically limited to ∼ 50 nm² and is indicative of arrested growth at the surface. Within a fully covered scan area, the ripening of the domains was found to be negligible even under the influence of continuous STM scanning. When using OA as a solvent, the experimental results obtained were similar in principle (see Fig. S2 In the Supplementary information). In both alkanoic acids, the polymerization-depolymerization process could be repeated multiple times in a given experimental session. The unit cell parameters of the 2DP and SAMN are provided in Table 1. Similar phase transitions have been reported in SAMNs of aromatic carboxylic acids[31–34]. Together with these previous reports, this work highlights the importance of the electric field in the manipulation of on-surface molecular self-assembly.

To ascertain if the acidic nature of the solvent and thus the availability of a dissociable proton is a critical factor in the polymerization and the subsequent bias-induced reversible depolymerization reaction, the on-surface synthesis was carried out at the MO–graphite interface. MO is structurally similar to OA except for the absence of the acidic carboxyl group. Figure 3a shows the STM image of the surface obtained at positive sample bias. Similar to the case of alkanoic acids, SAMN was formed under these conditions however oligomeric units of the boroxine 2DP were also formed in between the SAMN domains. The general morphology and unit cell parameters (Table 1) are identical to those observed in alkanoic acids. Furthermore, switching the sample bias to negative values led to a similar transition that yielded the 2DP as evident from Fig. 3b. The rate at which depolymerization proceeds in MO was found to be higher (see Fig. S3 in the Supplementary information) compared to that observed in the alkanoic acids, which is somewhat counterintuitive since the boroxine ring is expected to open under acidic conditions[35] and hence one would anticipate faster depolymerization in alkanoic acids compared to that in aprotic

**Table 1 | Unit cell parameters of the different types of networks formed by TPBA. For SAMN1, see Fig. S9 in the supplementary information**

| System | Unit cell parameters | | |
|---|---|---|---|
| | *a* (nm) | *b* (nm) | γ (°) |
| 2DP | 1.5 ± 0.1 | 1.5 ± 0.1 | 60.0 ± 1.0 |
| SAMN | 2.1 ± 0.1 | 2.1 ± 0.1 | 61.0 ± 1.0 |
| SAMN1 | 4.8 ± 0.1 | 4.8 ± 0.1 | 60.0 ± 1.0 |

**Fig. 3 | Bias-induced (de)polymerization of TPBA at the polar aprotic solvent-graphite interface.** Bias-induced (de)polymerization of TPBA at the MO–graphite (**a**, **b**) and TCB–graphite (**c**, **d**) interface. **a** SAMN formed at the MO–graphite interface with small islands of the 2DP (white arrows) in between the SAMN domains at positive sample bias. **b** 2DP formed at negative bias. **c** SAMN formed at the TCB–graphite interface at positive sample bias. **d** Co-existence of 2DP and SAMN at the TCB–graphite interface at negative sample bias. We hypothesize that the SAMN is formed on top of the 2DP which is adsorbed on the graphite surface. See also Fig. S4 in the supplementary information. Imaging conditions: $I_{set}$ = 0.1 nA, $V_{bias}$ = – 0.7 V or + 0.7 V. [TPBA] = 250 µM, Image size = 100 × 100 nm$^2$, scale bar = 20 nm.

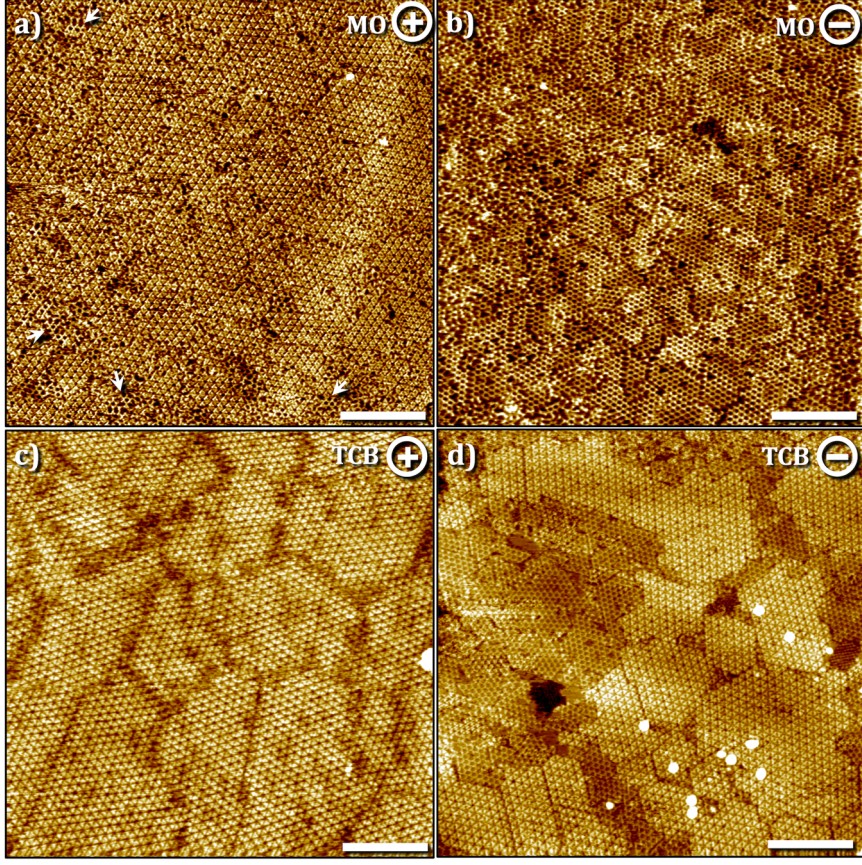

solvents. These observations further indicate that under these specific conditions, the acidity of the solvent does not have a significant influence on the depolymerization process and there may be other factors at play including the adsorption stability of the monomers under a given sample bias (vide infra).

Similar to the previous two cases, the formation of SAMN was observed at the TCB–graphite interface at positive bias (Fig. 3c). However, at negative bias, the coexistence of 2DP and SAMN was observed as evident from the STM image provided in Fig. 3d. A close inspection of the STM image indicates the plausible formation of a heterobilayer at the TCB–graphite interface where the bottom layer consists of the 2DP and a monolayer of SAMN is adsorbed on top of it. This hypothesis is not unreasonable considering the fact that the orientation of the unit cell of the 2DP in a given domain is often oriented at 4 ± 2° with respect to that of the SAMN indicating the templating effect of the former. We argue that if the two structures were co-adsorbed on the surface in separate and adjacent domains, one would not have expected this specific discrete relative orientation between the two. We note that the formation of such heterobilayer in the context of on-surface synthesized 2DPs has not been reported to date.

In contrast to the previous cases, the formation of the 2DP was not observed when 1-PO was used as a solvent, irrespective of the bias used for STM imaging. In 1-PO, only a self-assembled network with a different structure than that of SAMN was formed at negative bias which converted to an amorphous network at positive bias (Fig. S5 in the Supplementary information). This observation indicates that the nature of the solvent does have an impact on the bias-induced 2D (de)polymerization process. It also shows that the presence of residual amounts of DMSO, which is present in all solutions used in this study, is not a determining factor for the (de) polymerization process to occur and that the properties of the bulk solvent are more important. Furthermore, as mentioned earlier, bias-induced (de) polymerization in both HA and OA occurs identically, regardless of the presence of DMSO.

The role of residual amounts of water present in the solvent in the (de) polymerization process also merits some discussion here. The condensation of boronic acids releases water and hence it is possible to maintain equilibrium and keep the process reversible by regulating the amount of water present in the system. It has been demonstrated for boronic acid condensation carried out at the vapor solid interface that the presence of water regulating agents such as $CuSO_4.5H_2O$ in a closed reactor system leads to the formation of defect-free, extended domains whereas in the absence of such water "reservoir" only disordered domains of the 2DP were obtained[25]. In the present system, however, no water was added intentionally to regulate the equilibrium. In order to rule out the influence of water content present in HA and OA on the bias-induced (de)polymerization process, the STM experiments were carried out using dry solvents. These experiments yielded similar results to those obtained with solvents without drying. Furthermore, STM experiments were also carried out after the controlled addition of water to the STM solvents and the results of these experiments (Fig. S6 in the supplementary information) were comparable to those obtained with the anhydrous as well as undried solvents. This suggests that the amount of water present in these solvents does not significantly affect the polymerization/depolymerization process occurring at the solution–solid interface.

## EF mediated (de)polymerization: monomer concentration and time-dependence

Given the known dependence of 2DP formation on monomer concentration[18], we also investigated how the bias-induced (de)polymerization process is influenced by monomer concentration. For the solvent dependence of 2D polymerization described above, all observations were made for a monomer concentration of 250 µM. Here we describe in detail how the (de)polymerization process depends on the monomer concentration, especially how the dynamic adsorption and desorption of monomers can influence the reaction at the interface. For concentration-

**Fig. 4 | Time-dependence of depolymerization within and outside of the scanned area at the HA–graphite interface at [TPBA] = 250 μM.**
**a** Relatively large scale STM image showing domains of 2DP within the scanned area (100 × 100 nm², scale bar = 20 nm). **b** Smaller scan within the highlighted area in (**a**) immediately after switching the sample bias from negative to positive.
**b**–**g** Sequential STM images obtained in the same general area as (**a**) showing the local depolymerization process (50 × 50 nm², scale bar = 10 nm). **h** A larger scale image obtained after zooming out from (**g**) shows the locally depolymerized region (white square). The region outside of the scanned area still shows the presence of 2DP (100 × 100 nm², scale bar = 20 nm). Imaging conditions: $I_{set}$ = 0.1 nA, $V_{bias}$ = −0.7 V or +0.7 V. Similar local depolymerization was also observed when OA was used as the solvent (see Fig. S8 in the Supplementary information).

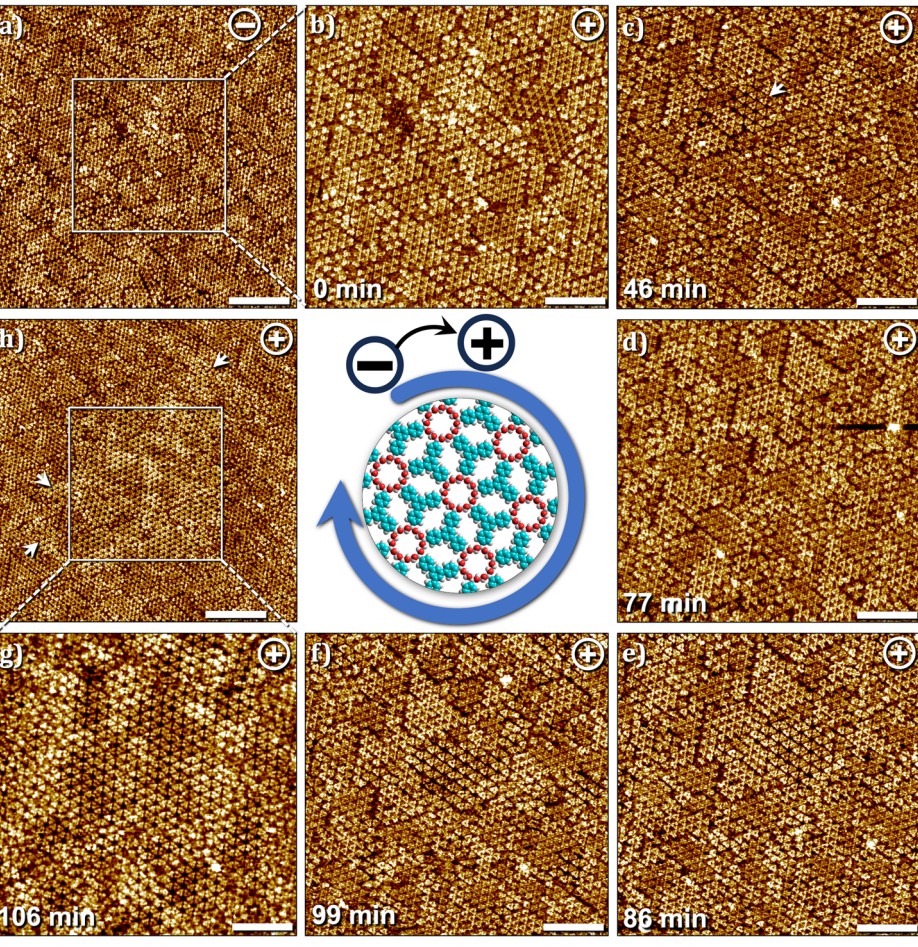

dependent experiments, the alkanoic acids were chosen as solvents, and a special focus was given to following the time-dependence of the (de)polymerization process within and outside of the scanned area.

Figure 4a shows the STM image of the graphite surface obtained after the deposition of a 250 μM solution of TPBA in HA at negative sample bias. As established earlier, under these experimental conditions, the surface is predominantly covered by a monolayer of the 2DP. Starting with this composition of the surface within the scanned area (100 × 100 nm²), the slow depolymerization process occurring in response to the change in the bias was followed by obtaining sequential STM images within approximately the same area after zooming in at 50 × 50 nm². Figure 4b shows the first scan immediately after switching the bias to positive and reveals that the 2DP domains remain virtually unchanged. Subsequent scans show that noticeable depolymerization and appearance of the SAMN do not begin until 4 min after the bias switch. Figure 4c shows a small domain of the SAMN (white arrow) which increases in size gradually over the next several minutes, and at the end of this period, the entire scanned area is covered by the SAMN (Fig. 4g). Upon zooming out back to 100 × 100 nm² revealed that the depolymerization has largely occurred within the scanned area and that the surrounding region still shows an abundance of 2DP domains (Fig. 4h), although a few SAMN domains were formed outside of the scanned area (white arrows, Fig. 4h). Further scanning of the larger area at positive bias resulted in the conversion of these 2DP domains into SAMN (Fig. S7 in the supplementary information).

In contrast to the rather slow depolymerization process described above, the polymerization to boroxine-linked 2DP was found to occur on a relatively faster timescale. The sequence presented in Fig. 5 shows time-dependent changes occurring in the surface adsorbed films at the HA–graphite interface in response to switching the bias from positive (predominantly SAMN and amorphous coverage of monomers) to negative (Fig. 5a→5b). Upon continuous scanning of the surface at negative polarity (Fig. 5b→5h), complete removal of the ordered SAMN and the consequent formation of the 2DP domains was observed within the scanned area. Since the nucleation of the 2DP occurs without any influence of the EF[20,23] (vide supra, see also Figure S1 in the supplementary information), we conclude that while the polymerization process is not initiated, and may or may not be accelerated by the EF at negative substrate bias, the depolymerization process, to a large extent, is initiated, as well as accelerated by the EF at positive substrate bias. As evident from the comparison of STM image sequences presented in Figs. 4 and 5, the depolymerization occurs at a much slower rate compared to polymerization at the heptanoic acid–graphite interface. Since the adsorption and desorption of monomers in response to the change in the polarity of the applied substrate bias is a spontaneous process (vide infra), one may conclude that the bond-breaking process determines the kinetics of the observed depolymerization process at the liquid-solid interface and that the cleavage of monomers from 2DP domains is the rate-limiting step.

At lower concentrations, the (de)polymerization process was found to be significantly different than that described above. Figure 6 shows a sequence of STM images where the influence of sample bias was studied by switching between positive and negative sample bias three times for a sample with monomer concentration of 25 μM. Starting at positive sample bias, the surface shows co-existence of SAMN, 2DP domains and amorphous arrangement of monomers (Fig. 6a). A bias switch to negative induced instantaneous desorption of the SAMN as well as disordered monomers present in the previous scan. Figure 6b clearly shows that only domains of 2DP remained on the surface together with a new structure

**Fig. 5 | Time-dependence of polymerization within the scanned area at the OA–graphite interface at [TPBA] = 250 µM.** Starting with a mixed composition of SAMN and 2DP in (**a**) this image sequence shows how the domains of SAMN are removed upon scanning at negative bias (**b**–**e**) with subsequent formation of 2DP (**f**–**h**). (50 × 50 nm$^2$, scale bar = 10 nm). Imaging conditions: $I_{set}$ = 0.1 nA, $V_{bias}$ = –0.7 V or +0.7 V.

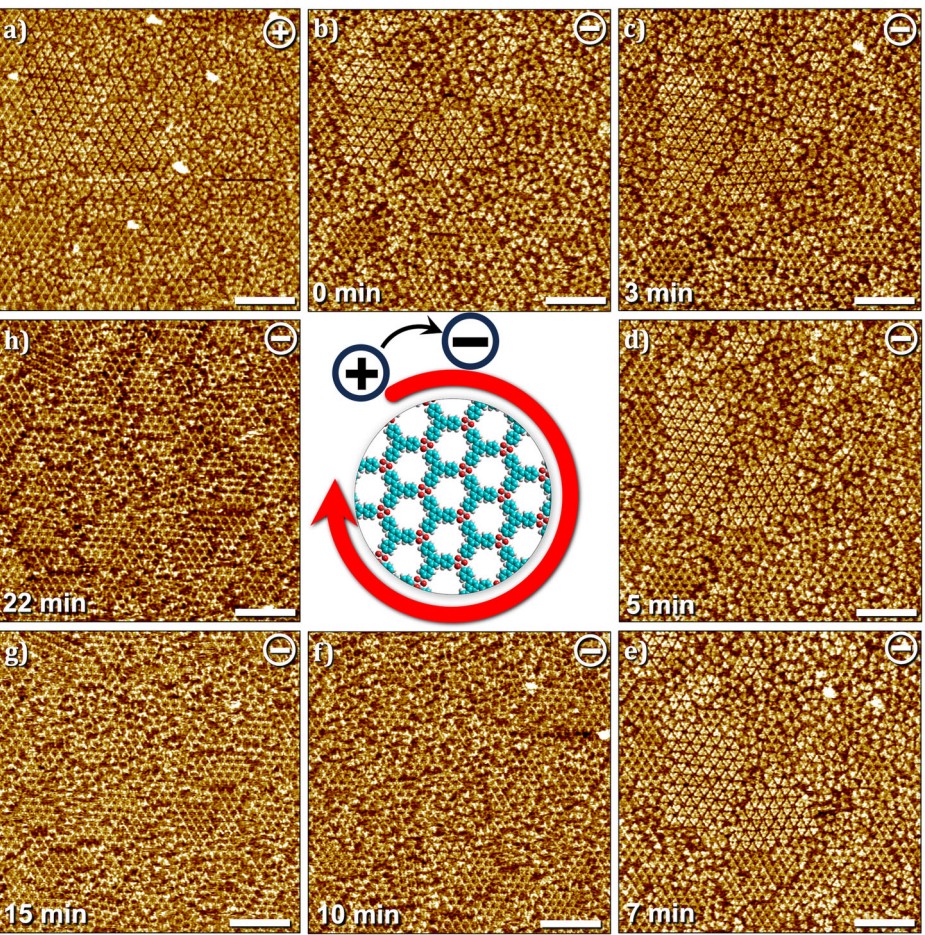

(SAMN1, see Table 1 for unit cell parameters) which we ascribe to the hydrogen-bonded assembly of covalently linked trimers of TPBA. For additional STM images of this new phase and the proposed molecular model, see Fig. S9 in the supplementary information. Such partial covalent structures have been reported earlier for boroxine-based 2DPs[18]. The assembly of covalent trimers is also present at positive bias (Fig. S10 in the supplementary information) however cannot be discerned clearly due to the adsorption of a TPBA monomer inside the central hexagonal cavity. At negative bias, all monomers, which are part of the SAMN, adsorbed as amorphous matrix and as guests within the hydrogen-bonded trimeric assembly are desorbed from the surface (see for example, the area enclosed by the white dashed line, Fig. 6a, b). Reversing the sample bias back to positive led to instant re-adsorption of the monomers on the surface (Fig. 6c). The re-adsorbed monomers however mostly form an amorphous matrix with a few nuclei reminiscent of the SAMN (area enclosed by blue lines, Fig. 6d). These nuclei evolved into ordered domains of SAMN in subsequent scans when scanning was continued at positive bias (Fig. 6c–e). This sequence can be repeated wherein negative bias promotes the desorption of the monomers (Fig. 6e, f) and positive bias promotes their re-adsorption (Fig. 6g, h).

The bias-dependent reversible adsorption/desorption of boronic acid monomers was recently rationalized using ultraviolet photoemission spectroscopy (UPS)[23]. UPS revealed that the work function of graphite is reduced upon adsorption of the boronic acid monomers which is indicative of electron transfer from the monomers to the graphite surface. It was thus argued that the application of positive substrate bias, wherein the electrons tunnel from the tip to the substrate, facilitates the electron transfer from the monomers to the graphite substrate thereby promoting

their adsorption. The change in the bias, which essentially changes the direction of electron tunneling was proposed to have the opposite effect, thus disfavoring the adsorption of monomers on the surface. The observed changes could thus be hypothesized to be occurring due to the known concentration dependence of on-surface assembly[36–38] and 2D polymerization[17,18,39]. A denser structure, (here the SAMN) is favored at higher (local) concentration (at positive substrate bias) and a porous network (here the 2DP) is formed when the (local) monomer concentration is reduced in response to desorption of the monomers at the negative bias[23]. Similar considerations may apply to the current system as well. In fact, continuous scanning of the substrate with partial surface coverage of the 2DP at negative substrate bias did not lead to an appreciable increase in the surface coverage of the 2DP domains (see Fig. S13 and Supplementary Video 1) further confirming that the polymerization itself is not accelerated by the negative substrate bias and could indeed be linked to the local concentration of monomers.

Figure 7 shows a sequence of STM images, where starting with a partial surface coverage of the 2DP at negative sample bias (Fig. 7a), the polarity of the voltage was switched to positive and the resulting changes in the surface were monitored. As anticipated, the bias switch immediately promotes the adsorption of TPBA monomers. The first scan after the switch shows ill-defined regions in between the 2DP domains (Fig. 7b) which begin to get ordered in subsequent scans. This adsorption and assembly of the monomers is accompanied by a simultaneous reduction in the surface coverage of the 2DP domains. The domains of 2DP highlighted in dotted triangles provide a guide to the eye (Fig. 7a–f) and reveal how the area in between the domains, which was previously occupied by the 2DP gets covered with domains of SAMN at positive substrate bias. The change in the position of

**Fig. 6 | Adsorption–desorption dynamics of TPBA monomers at the HA–graphite interface.**
**a–h** Sequential STM images obtained in approximately the same area showing the instantaneous and reversible desorption and adsorption of TPBA monomers at the HA–graphite interface. As evident from the sequence, desorption of monomers occurs at negative bias while the domains of 2DP and the network of hydrogen-bonded trimers remain on the surface (white dashed lines, **a→b**, **e→f**). On the other hand, a change in the sample bias from negative to positive leads to instantaneous re-adsorption of the monomers (**b→c**, **g→h**), although the monomer assembly is relatively amorphous. This amorphous structure transforms into ordered domains if the scanning is continued at positive sample bias (area highlighted in blue, **c→d→e**). (100 × 100 nm², scale bar = 20 nm). Imaging conditions: $I_{set}$ = 0.1 nA, $V_{bias}$ = − 0.7 V or + 0.7 V. [TPBA] = 25 µM. (see also Fig. S11 in the supplementary information).

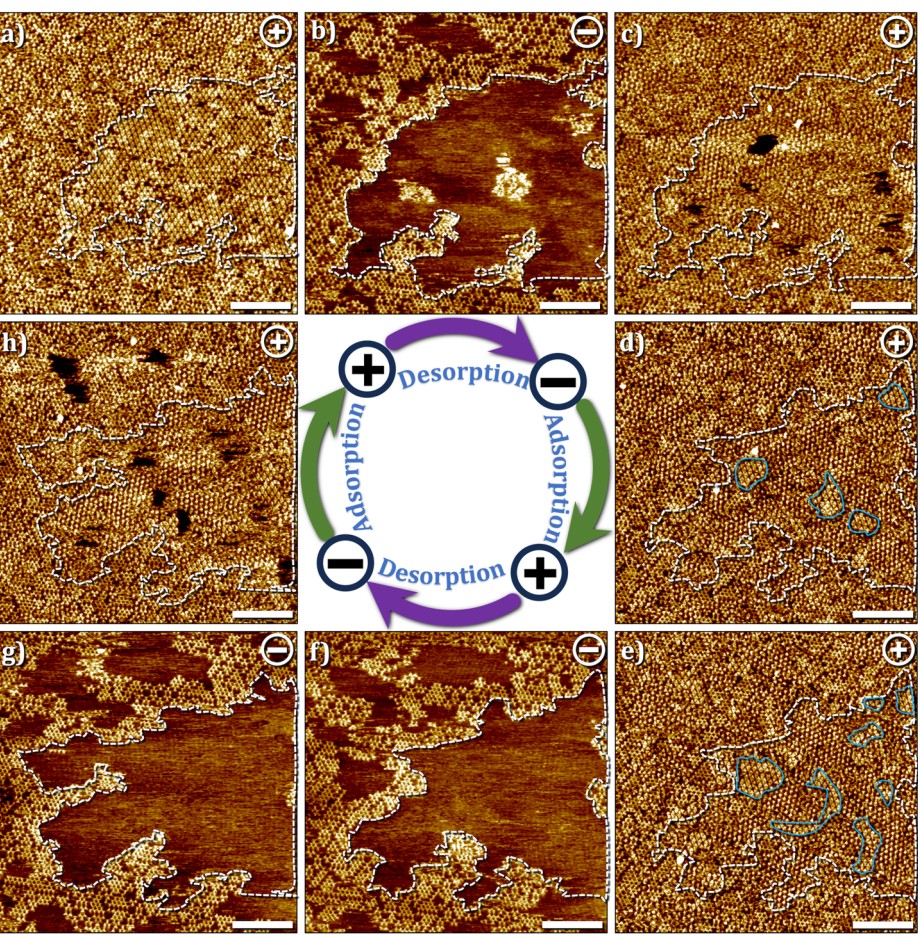

the highlighted domains in sequential STM images is due to the thermal drift of the STM scanner which moves the scanned area gradually to the left. At this juncture, we cannot clearly separate the "desorption" process of the 2DP domains from their on-surface "depolymerization". We hypothesize that the two processes are coupled and that the gradual depolymerization leaves smaller and smaller fragments of the 2DP on the surface in subsequent scans at positive bias which are eventually desorbed from the surface with the concomitant increase in the surface coverage of the SAMN domains.

## Conclusions and outlook

Locally applied strong electric fields have the potential to be used as smart reagents for controlling chemical reactions. On-surface synthesis of 2DPs and their molecular scale characterization using STM provides an intriguing test bed for understanding the influence of local electric fields on the reactivity and structure of molecules and assemblies, respectively. Building on the previous reports, we have described above how the nature of the solvent, and the concentration of monomers affect the EF-mediated polymerization and depolymerization of a boroxine-based system. While the polymerization, as well as EF-mediated transition between SAMN and 2DP, could be achieved in polar protic as well as polar aprotic solvents, only SAMN formation was observed in a non-polar solvent. Our results indicate that the nucleation of small domains of 2DP occurs even in the absence of an electric field as also reported recently by others. The subsequent growth of the 2DP is plausibly aided by STM scanning at negative bias. Based on the experiments carried out at lower monomer concentrations, we conclude that the depolymerization and the subsequent desorption of the 2DP domains are initiated and accelerated when the substrate bias is switched to positive.

The experimental results clearly indicate that the adsorption (desorption) of monomers and the depolymerization/desorption of 2DP (adsorption) are coupled. Given the known concentration dependence of on-surface assemblies, it is not unreasonable to conclude that the observed transition is a result of preferential adsorption of a concentration-controlled structure at the solution–solid interface. The fact that the net result of the bias-dependent transition, namely 2DP formation at negative substrate bias and SAMN formation at positive substrate bias, remains always the same irrespective of the total solution concentration, also confirms that the local concentration changes occurring at the solution–solid interface drive the observed changes in the surface structure instead of the solution concentration itself. The ability to initiate and control polymerization and depolymerization at will using change in the electric field offers an interesting test bed for studying the nucleation, growth, and ripening phenomena transpiring during on-surface 2D polymerization.

While the results described above certainly advance our understanding of the EF-mediated 2D (de)polymerization occurring at the solution–solid interface, several aspects still need detailed scrutiny. The adsorption of monomers at positive substrate bias correlates with the gradual disappearance of the 2DP domains, however, it is not certain if the former is responsible for the depolymerization itself. Additionally, the role of water needs to be investigated further as it is virtually impossible to remove traces of water from the solvent which is eventually used under ambient conditions. One must also bear in mind that even when working under perfectly anhydrous conditions, water is released as a by-product of the condensation process and its fate under strong electric fields within an organic environment is a complex aspect and may contribute to the processes transpiring during the EF-mediated transitions. Further attempts to understand some of these aspects both via experiment and theory are warranted.

**Fig. 7 | Depolymerization dynamics at the HA–graphite interface. a–h** Sequential STM images showing the gradual removal of the 2DP domains from the surface of graphite and the concomitant adsorption of TPBA monomers and growth of SAMN domains at the HA–graphite interface. As evident from the image sequence, the 2DP domains between those marked by colored triangles gradually shrink and are removed from the surface. The position of marked triangles shifts in subsequent images due to the drift of the scanner which is larger in this sequence. Imaging conditions: $I_{set}$ = 0.1 nA, $V_{bias}$ = − 0.7 V or + 0.7 V. [TPBA] = 25 µM. (see also Fig. S12 in the supplementary information).

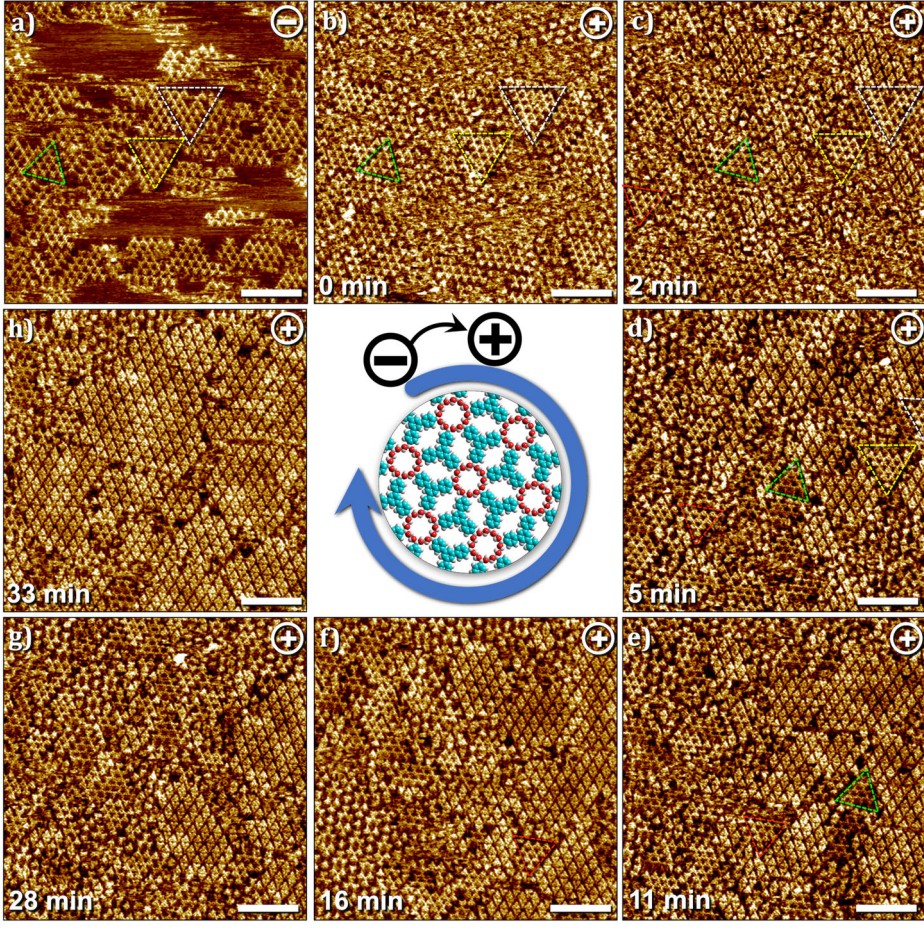

## Methods

### Materials and sample preparation

1,3,5-Tris(4-phenylboronic acid) benzene (BLDpharm, 98%), heptanoic acid (Sigma-Aldrich, ≥99%), octanoic acid (Sigma-Aldrich, ≥ 99%), methyl octanoate (TCI, >99%), 1,2,4-trichlorobenzene (Sigma-Aldrich, ≥99%), 1-phenyloctane(Thermo scientific, 99%) and dimethylsulfoxide (Sigma-Aldrich, ≥99%) were used directly without further purification. Solutions of 1,3,5-tris(4-phenylboronic acid) benzene were prepared by dissolving the solid compound in dimethylsulfoxide at a ratio of 1 mg/ml. The DMSO stock solution was further diluted using a specific solvent listed above to generate a concentration series (the amount of DMSO in each solution is less than 1.1%V/V) for the STM experiments. For a few STM experiments, OA was dried by stirring over freshly activated molecular sieves (Carl Rot) for 48 h and was stored in a round bottom flask over anhydrous sodium sulfate.

### STM experiments and image processing

All STM experiments were conducted using a PicoSPM (Agilent) machine operating in constant-current mode at room temperature. STM tips were prepared by mechanically cutting a Pt/Ir wire (80/20 alloy, diameter 0.2 mm, Advent Research Materials). HOPG (grade ZYB, Momentive Performance Material Quartz Inc., Strongsville, OH, USA) was utilized as the substrate for STM measurements at the liquid-solid interface under ambient conditions. Multiple samples were investigated, and for each sample, several locations were probed. The bias voltage refers to the substrate bias. For analysis purposes, the recording of a monolayer image was followed by imaging the graphite substrate underneath it under the same experimental conditions, except for increasing the current and lowering the bias. The images were corrected for drift via Scanning Probe Image Processor (SPIP) software (Image Metrology ApS), using the recorded graphite images for calibration purposes, allowing a more accurate unit cell determination. The unit cell parameters were determined by examining at least four images and only the average values are reported. All images are Gaussian filtered. The STM images in Fig. 1 are correlation averaged. Imaging parameters for the STM images are indicated in the figure captions and labeled as $V_{bias}$ for sample bias and $I_{set}$ for tunneling current. The molecular models were constructed using the HyperChem program.

## Data availability

The experimental data underlying this study are openly available in KU Leuven Research Data Repository at https://doi.org/10.48804/BXGCWO.

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

## Acknowledgements

We thank Dr. Niklas Herrmann for drying the octanoic acid, and for reading the manuscript and providing feedback. Financial support from the Research Foundation—Flanders (FWO) (grant G0A4120N) and KU Leuven–Internal Funds (C14/19/079 and C14/23/090) is acknowledged. This work was in part supported by FWO (Research Foundation—Flanders (FWO)) and F.R.S.-FNRS under the Excellence of Science EOS program (projects 30489208 and 40007495). The research received in part also funding from the European Union under the Horizon Europe grant 101046231 (FantastiCOF) and M-ERA.NET via FWO (G0K9822N) (Super-Super). N.A. acknowledges a postdoctoral fellowship from (FWO) (grant 12ZS623N). Z.F. acknowledges the China Scholarship Council (CSC) (File no. 202006230117).

## Author contributions

K.S.M. and S.D.F. conceived of the project. Z.F. performed all experiments and collected and processed the data. N.A. supervised the experiments. Z.F., K.S.M., N.A., and S.D.F. co-wrote the manuscript.

## Competing interests

The authors declare no competing interests.
