## [Peer Review File · Communications Chemistry]

Reviewers' comments:

Reviewer #1 (Remarks to the Author):

In the present manuscript, Zhinan Fu and co-workers present a comprehensive and insightful investigation into the electric field (EF)-mediated polymerization and de-polymerization processes of boroxine-based 2D polymer at the solution-solid interface. The authors systematically explore the impact of solvent choice on EF-induced transformations, providing a thorough investigation supported by scanning tunneling microscopy experiments in both protic and aprotic solvents. The oscillatory behavior observed at higher concentrations, leading to the formation of ordered self-assembled molecular networks (SAMNs) and crystalline 2DP domains upon polarity switch, presents an intriguing aspect of the study. The conclusions are well-supported by the data, which is of high quality. With an impressive set of STM experiments, the authors have acquired insight into the structure and the dynamics at play. The manuscript is well-written, and the illustrations are instructive and appealing. I would recommend the minor revisions by addressing the following comments:

Major comments:

1. A transition from positive to negative bias leads to polymerization at the heptanoic acid/graphite interface. Can the authors discuss the significance of this bias-dependent transition and elaborate on any observed time-dependent aspects during this transition?
2. The time-dependent changes in the depolymerization process at the heptanoic acid (HA)/graphite interface in response to bias switching. Can the authors elaborate on the observed differences in the timescales of depolymerization and polymerization processes? Additionally, what implications do these contrasting timescales have on the understanding of the kinetics and dynamics of the depolymerization reactions, especially concerning the role of the electric field?
3. The polymerization-depolymerization process can be repeated multiple times in a given experimental session in both heptanoic acid and octanoic acid solvents. Can the authors elaborate on the factors or mechanisms that allow for the reproducibility of this process, and discuss any observed variations or limitations in the successive cycles of depolymerization?
4. I would suggest the authors to provide insights on other 2D polymers, including those based on imine and imide, which are widely studied. The described chemistry of polymerization and depolymerization in boroxine-based 2D polymers in the presented work would be expected to be similar to the chemistry of polymerization and depolymerization in other imine and imide bonds.

Minor comment:

Please cross-check references 25 and 34; '2d' should be corrected to '2D'.

Reviewer #2 (Remarks to the Author):

This is an interesting work reporting the polymerization and depolymerization on HOPG/solvent interfaces influenced by applied electric fields. Carefully STM studies provide an intriguing test for understanding the influence of local electric fields on the reactivity and structure of molecular assemblies. Although on-surface synthesis of 2DP of the same system has been reported in the previous works (partially by the same group), the electrically field induced reactions were rarely explored thus brought new aspects for this research field. I think the work can be acceptable for publication in *Communication Chemistry*, after the following points are addressed:

1. What I miss mostly is the mechanism for depolymerization under negative bias. The authors

claimed that the nucleation of 2DP domains is not initiated by the applied electric field, but the depolymerization and subsequent desorption is a consequence of the change in the polarity. In this context, I would expect more mechanical explanation for the electrical field induced depolymerization rather than just adsorption/desorption consideration. Are the 2DP decomposed under electrical field or not?

2. The group has done the thermal induced 2DP with the same monomer. Are these 2DPs becoming depolymerized under positive bias?

Some typos need to be checked. E.g. line 138: SAMN is in bold, others are not; line 243 ",The..." should be corrected.

Reviewer #3 (Remarks to the Author):

De Feyter et al. discussed the factors influencing electric field mediated (de)polymerization using STM and the molecular system of TPBA. Systematic discussions together with experiments carried out were presented in a well-organized manner. The writing skill for scientific paper is on the top level and thus I like this manuscript very much.

After reading, I realize that electric field may not promote the occurrence of such polymerizations. By contrast, the polarity of STM can lead to molecular desorption and adsorption (as concluded by this work). Along this thinking, it is important to highlight the significance of the so-called "STM-induced" molecular self-assembling or close packing (also the concept of proximity for surface synthesis), which may be the key to the success of the surface synthesis via STM.

Reviewer #4 (Remarks to the Author):

This work investigates the mechanism of EF induced 2D polymerization of aryl boronic acid. The EF effect had been reported by the same group. This manuscript presents detailed investigation the effect of different factors, including solvent, concentration, on the polymerization/depolymerization process. The reversible transition is proposed to be related to the bias-induced adsorption and desorption of the monomers, which drive changes in the local concentration of the monomers. The experiments are well-designed and the deduction process is convincing. The paper is suitable for Communication Chemistry.

1. Please provide a brief discussion of the orientation of different domains of 2DP and SAMN.

Reviewers' comments:

Reviewer #1 (Remarks to the Author):

In the present manuscript, Zhinan Fu and co-workers present a comprehensive and insightful investigation into the electric field (EF)-mediated polymerization and de-polymerization processes of boroxine-based 2D polymer at the solution-solid interface. The authors systematically explore the impact of solvent choice on EF-induced transformations, providing a thorough investigation supported by scanning tunneling microscopy experiments in both protic and aprotic solvents. The oscillatory behavior observed at higher concentrations, leading to the formation of ordered self-assembled molecular networks (SAMNs) and crystalline 2DP domains upon polarity switch, presents an intriguing aspect of the study. The conclusions are well-supported by the data, which is of high quality. With an impressive set of STM experiments, the authors have acquired insight into the structure and the dynamics at play. The manuscript is well-written, and the illustrations are instructive and appealing. I would recommend the minor revisions by addressing the following comments:

Major comments:

1. A transition from positive to negative bias leads to polymerization at the heptanoic acid/graphite interface. Can the authors discuss the significance of this bias-dependent transition and elaborate on any observed time-dependent aspects during this transition?

Author response:

We would like to emphasize that when starting with a surface covered by the self-assembled molecular network (SAMN, imaged at a positive bias), switching to the negative polarity results in the formation of 2D polymer domains on the surface. However, as stated in the manuscript, and also reported in *Ref. 23*, the presence of an electric field at negative substrate bias is not a necessary condition for the formation of the 2D polymers at the alkanoic acid/graphite interface. We conclude this based on the observation that the first STM scan of an experimental session also shows domains of 2D polymer on the surface indicating spontaneous 2D polymerization at the solution-solid interface. The polymerization is plausibly accelerated by the strong electric field applied at negative substrate bias. The ability to depolymerize and polymerize using the electric field as a trigger is an interesting and ingenious strategy for studying the 2D polymerization process since it allows monitoring of nucleation, growth, and/or ripening processes in real-time at the liquid-solid interface.

To highlight the importance of the bias dependence transition reported in our manuscript, we have added the following sentence to the revised version of the manuscript.

“The ability to initiate and control polymerization and depolymerization at will using change in the electric field offers an interesting test bed for studying the nucleation, growth, and ripening phenomena transpiring during on-surface 2D polymerization.”

The time-dependence of the phase transitions was already discussed in detail in the original version of the manuscript. Since the time dependence was discussed together with the concentration dependence (Figure 3-Figure 6), it was probably not clearly visible. To make it

clear to the readers, we have changed a section title to also include the time-dependent aspects of the process.

We have modified the section title from “EF mediated (de)polymerization: Influence of monomer concentration” to “EF mediated (de)polymerization: Monomer concentration and time-dependence”

2. The time-dependent changes in the depolymerization process at the heptanoic acid (HA)/graphite interface in response to bias switching. Can the authors elaborate on the observed differences in the timescales of depolymerization and polymerization processes? Additionally, what implications do these contrasting timescales have on the understanding of the kinetics and dynamics of the depolymerization reactions, especially concerning the role of the electric field?

Author response:

As presented in Figures 3 and 4, at a constant monomer concentration, the time taken for depolymerization (Figure 3, 106 minutes) is much slower than it takes for polymerization (Figure 4, 22 minutes). The former transition occurs upon switching the polarity from negative to positive bias whereas the latter occurs upon positive to negative polarity switch. We have demonstrated in Figure 6 that the adsorption and the desorption of the monomers is a spontaneous process. This means that depolymerization is a much slower process compared to polymerization since the rate of adsorption as well as desorption of free monomers is fast. Furthermore, as stated in the manuscript (also see the response to the comment above), polymerization to some extent is a spontaneous process whereas the depolymerization within the scanned area is only triggered by the change in the polarity of the applied electric field. All these factors taken together explain the difference in the time scales of the two processes. Based on these considerations, one can also conclude that the bond-breaking process determines the kinetics of the observed depolymerization process at the liquid-solid interface and that the cleavage of monomers from 2DP domains is the rate-limiting step.

To clarify this point further, we have added the following discussion to the revised version of the manuscript:

“As evident from the comparison of STM image sequences presented in Figures 3 and 4, the depolymerization occurs at a much slower rate compared to polymerization at the heptanoic acid/graphite interface. Since the adsorption and desorption of monomers in response to the change in the polarity of the applied substrate bias is a spontaneous process (vide infra, Figure 6), one may conclude that the bond-breaking process determines the kinetics of the observed depolymerization process at the liquid-solid interface and that the cleavage of monomers from 2DP domains is the rate-limiting step.”

3. The polymerization-depolymerization process can be repeated multiple times in a given experimental session in both heptanoic acid and octanoic acid solvents. Can the authors elaborate on the factors or mechanisms that allow for the reproducibility of this process, and discuss any observed variations or limitations in the successive cycles of depolymerization?

Author response:

We thank the reviewer for pointing out this important aspect. We have indeed observed some session-to-session variations in the time scale of the polymerization and depolymerization process. For example, the time required for bias-dependent polymerization while scanning an area of 50 nm × 50 nm differed by up to 50% in two different sessions. This was observed for both octanoic as well as heptanoic acid. Similar differences were observed for the depolymerization process as well. These differences could be related to the size of the domains of the SAMN (or the 2DP) at the point of bias-switch and the overall stability of the STM feedback. One expects a larger domain to be more stable and thus take longer to depolymerize/desorb at the liquid-solid interface. We have included this information in the revised supporting information. We would like to emphasize that while there is variation in the time scales from session to session, in general, the depolymerization process was always found to be slower than the 2D polymerization process.

To clarify this point further, we have added the following text to the revised version of the supporting information.

“Some session-to-session variations in the time scale of the polymerization and depolymerization process were observed. For example, the time required for bias-dependent polymerization while scanning an area of 50 nm × 50 nm differed by up to 50% in two different sessions. This was observed for both octanoic as well as heptanoic acid. Similar differences were observed for the depolymerization process as well. These differences could be possibly related to the size of the domains of the SAMN (or the 2DP) at the point of bias-switch and the overall stability of the STM feedback. One expects a larger domain to be more stable and thus take longer to depolymerize/desorb at the liquid-solid interface. We emphasize that while there is variation in the time scales from session to session, in general, the depolymerization process was always found to be slower than the 2D polymerization process.”

4. I would suggest the authors to provide insights on other 2D polymers, including those based on imine and imide, which are widely studied. The described chemistry of polymerization and depolymerization in boroxine-based 2D polymers in the presented work would be expected to be similar to the chemistry of polymerization and depolymerization in other imine and imide bonds.

Author response:

While the on-surface synthesis of imine-based 2D polymers and their characterization at the liquid-solid interface using STM has been reported, there exist no such reports for the imide-based 2D polymers. As of now, the electric field-induced polymerization and depolymerization process has only been studied for boroxine- and boronate ester-based 2D polymers. The influence of changes in the polarity of the substrate bias for surface synthesized imine 2DPs has not been reported to date. While there is the possibility that similar processes might occur, given the complex nature of the EF-mediated (de)polymerization process, we refrain from speculating on these systems which merit a thorough investigation.

Minor comment:

Please cross-check references 25 and 34; '2d' should be corrected to '2D'.

Author response:

We thank the reviewer for pointing out this typo. It has been corrected in the revised manuscript.

Reviewer #2 (Remarks to the Author):

This is an interesting work reporting the polymerization and depolymerization of HOPG/solvent interfaces influenced by applied electric fields. Carefully STM studies provide an intriguing test for understanding the influence of local electric fields on the reactivity and structure of molecular assemblies. Although on-surface synthesis of 2DP of the same system has been reported in the previous works (partially by the same group), the electrically field-induced reactions were rarely explored thus bringing new aspects to this research field. I think the work can be acceptable for publication in Communication Chemistry after the following points are addressed:

1. What I miss mostly is the mechanism for depolymerization under negative bias. The authors claimed that the nucleation of 2DP domains is not initiated by the applied electric field, but the depolymerization and subsequent desorption is a consequence of the change in the polarity. In this context, I would expect a more mechanical explanation for the electrical field-induced depolymerization rather than just adsorption/desorption consideration. Is the 2DP decomposed under an electric field or not?

Author response:

Based on the STM data presented in Figure 6, we can conclude that the 2DP is indeed decomposed at positive substrate bias. In the STM images provided in Figures 6a and 6b, it can be readily noticed that a part of the 2D polymer domain adjacent to the white dotted triangle undergoes size reduction which is indicative of on-surface depolymerization. However, given the slow nature of STM scanning (one to two minutes for the typical scan sizes used in the current work), we do not wish to exclude possible desorption events that may be coupled to depolymerization. Hence, we have always maintained throughout the manuscript that the two processes may be coupled.

2. The group has done the thermal-induced 2DP with the same monomer. Are these 2DPs becoming depolymerized under positive bias?

Author response:

We have indeed previously reported on the *ex-situ* synthesized 2DP based on boroxines. Unfortunately, no bias-dependent measurements were carried out for that study (*Chem. Commun.* **2016**, 52, 68-71). However, we expect that the thermally synthesized polymer will also undergo depolymerization if an appropriate solvent is present in the system. In our previous work with a larger monomer, we have concluded that depolymerization does not proceed on dry samples (*J. Am. Chem. Soc.* **2019**, 141, 11404–11408) and the presence of an octanoic acid was necessary for the process to occur.

Some typos need to be checked. E.g. line 138: SAMN is in bold, others are not; line 243 “,The...” should be corrected.

Author response:

We thank the reviewer for pointing out these typos. They have been corrected in the revised manuscript.

Reviewer #3 (Remarks to the Author):

De Feyter et al. discussed the factors influencing electric field mediated (de)polymerization using STM and the molecular system of TPBA. Systematic discussions together with experiments carried out were presented in a well-organized manner. The writing skill for scientific paper is on the top level and thus I like this manuscript very much.

We thank the reviewer for the positive evaluation of our manuscript.

1. After reading, I realized that an electric field may not promote the occurrence of such polymerizations. By contrast, the polarity of STM can lead to molecular desorption and adsorption (as concluded by this work). Along with this thinking, it is important to highlight the significance of the so-called “STM-induced” molecular self-assembling or close packing (also the concept of proximity for surface synthesis), which may be the key to the success of surface synthesis *via* STM.

Author response:

We thank the reviewer for pointing out this aspect. We have added the following sentences in the revised version of the manuscript:

“Similar phase transitions have been reported in SAMNs of aromatic carboxylic acids.^{31, 32, 33, 34} Together with these previous reports, this work highlights the importance of the electric field in the manipulation of on-surface molecular self-assembly.”

“Given the proximity of a solid surface, the reactions often proceed under relatively mild conditions.”

Reviewer #4 (Remarks to the Author):

This work investigates the mechanism of EF-induced 2D polymerization of aryl boronic acid. The EF effect had been reported by the same group. This manuscript presents a detailed investigation of the effect of different factors, including solvent, and concentration, on the polymerization/depolymerization process. The reversible transition is proposed to be related to the bias-induced adsorption and desorption of the monomers, which drive changes in the local

concentration of the monomers. The experiments are well-designed and the deduction process is convincing. The paper is suitable for Communication Chemistry.

We thank the reviewer for the positive feedback on the manuscript.

1. Please provide a brief discussion of the orientation of different domains of 2DP and SAMN.

Author response:

The unit cell vectors of the 2DP domains were found to be coincident with the symmetry axes of graphite lattice. The unit cell of the SAMN1 network was oriented at a small ($\sim 3^\circ$) angle with respect to the symmetry axes of the underlying graphite lattice. Lastly, the SAMN unit cell was rotated by $\sim 8^\circ$ with respect to one of the symmetry axes of the graphite lattice. This information has been added to the revised supporting information (Figure S17, see also below).

Figure S17. STM images calibrated using the graphite lattice showing the orientation of 2DP, SAMN1 (a), and SAMN (b) with respect to the symmetry axes of graphite. The unit cell vectors of the 2DP domains were found to be coincident with the symmetry axes of graphite lattice. The unit cell of the SAMN1 network was oriented at a small ($\sim 3^\circ$) angle with respect to the symmetry axes of the underlying graphite lattice. Lastly, the SAMN unit cell was rotated by $\sim 8^\circ$ with respect to one of the symmetry axes of the graphite lattice.

REVIEWERS' COMMENTS:

Reviewer #1 (Remarks to the Author):

I am pleased with the revisions. The revised manuscript is well-structured, and its conclusions are supported by high-quality data. The impressive set of STM experiments provides valuable insights into the structure and dynamics. Therefore, I highly recommend the publication of manuscript in its current form.

Reviewer #2 (Remarks to the Author):

The authors have addressed all my questions. I would recommend for publication without further revisions.